# Lignin-Loaded Carbon Nanoparticles as a Promising Control Agent against *Fusarium verticillioides* in Maize: Physiological and Biochemical Analyses

**DOI:** 10.3390/polym15051193

**Published:** 2023-02-27

**Authors:** Sherif Mohamed El-Ganainy, Mohamed A. Mosa, Ahmed Mahmoud Ismail, Ashraf E. Khalil

**Affiliations:** 1Department of Arid Land Agriculture, College of Agricultural and Food Sciences, King Faisal University, P.O. Box 420, Al-Ahsa 31982, Saudi Arabia; 2Pests and Plant Diseases Unit, College of Agricultural and Food Sciences, King Faisal University, P.O. Box 420, Al-Ahsa 31982, Saudi Arabia; 3Vegetable Diseases Research Department, Plant Pathology Research Institute, Agricultural Research Center (ARC), Giza 12619, Egypt; 4Nanotechnology & Advanced Nano-Materials Laboratory (NANML), Plant Pathology Research Institute, Agricultural Research Center, Giza 12619, Egypt; 5Nematology Research Department, Plant Pathology Research Institute, Agricultural Research Center, Giza 12619, Egypt

**Keywords:** lignin-loaded carbon nanoparticles, seed coating, antifungal, *Zea mays*, *Fusarium verticillioides*

## Abstract

Lignin, a naturally occurring biopolymer, is produced primarily as a waste product by the pulp and paper industries and burned to produce electricity. Lignin-based nano- and microcarriers found in plants are promising biodegradable drug delivery platforms. Here, we highlight a few characteristics of a potential antifungal nanocomposite consisting of carbon nanoparticles (C-NPs) with a defined size and shape containing lignin nanoparticles (L-NPs). Spectroscopic and microscopic studies verified that the lignin-loaded carbon nanoparticles (L-CNPs) were successfully prepared. Under in vitro and in vivo conditions, the antifungal activity of L-CNPs at various doses was effectively tested against a wild strain of *F. verticillioides* that causes maize stalk rot disease. In comparison to the commercial fungicide, Ridomil Gold SL (2%), L-CNPs introduced beneficial effects in the earliest stages of maize development (seed germination and radicle length). Additionally, L-CNP treatments promoted positive effects on maize seedlings, with a significant increment in the level of carotenoid, anthocyanin, and chlorophyll pigments for particular treatments. Finally, the soluble protein content displayed a favorable trend in response to particular dosages. Most importantly, treatments with L-CNPs at 100 and 500 mg/L significantly reduced stalk rot disease by 86% and 81%, respectively, compared to treatments with the chemical fungicide, which reduced the disease by 79%. These consequences are substantial considering the essential cellular function carried out by these special natural-based compounds. Finally, the intravenous L-CNPs treatments in both male and female mice that affected the clinical applications and toxicological assessments are explained. The results of this study suggest that L-CNPs are of high interest as biodegradable delivery vehicles and can be used to stimulate favorable biological responses in maize when administered in the recommended dosages, contributing to the idea of agro-nanotechnology by demonstrating their unique qualities as a cost-effective alternative compared to conventional commercial fungicides and environmentally benign nanopesticides for long-term plant protection.

## 1. Introduction

Maize (*Zea mays* L.) is the most cultivated and consumed grain worldwide [1]. It is a diverse crop with over 600 derivative products that have a range of uses in agriculture (livestock and poultry) and the energy sector [2]. The Food and Agricultural Organization (FAO) reported that the global goal of (2.4%) production yield improvement annually would not be met because estimates predict a (7%) yield decline annually due to biotic and abiotic factors [3,4,5], among which diseases caused by *F. verticillioides*. This fungus takes advantage of the crop’s weakened defense system to flourish in a hot and dry environment and then enters maize tissues through injuries caused by insects, seeds, and silks [6,7]. Following this, disease propagules are transmitted from seeds to kernels in four primary stages: (i) seeds to seedlings, (ii) stalk colonization, (iii) movement into the ear, and (iv) spread within the ear, which results in significant economic impacts [8,9]. Different management strategies have recently been applied to reduce disease incidence [10,11]. Chemical pesticides represent the most effective solution against fungal disease [12]. However, the long-term use of these agrochemicals has induced undesirable pathogen resistance [13]. In addition, their residues in soil and food are detrimental to human beings and the environment [14]. Therefore, developing efficient and secure substitutes to manage *F. verticilloides* is urgently needed. 

In the past several years, agriculture has adopted nanotechnology as a novel alternative strategy for controlling plant diseases and as a viable platform for the more sustainable release of active ingredients, such as nanofertilizers [15], nanopesticides [16], and genetic engineering [17]. Dispersions, quantum dots, metal oxide nanoparticles, and nanocarriers (NCs) made of silica, lipids, polymers, or organic materials are the most common nanostructures used in precision agriculture [18,19]. In several areas of research, carbon-based nanomaterials (CNMs) are being investigated as drug carrier vehicles and smart delivery systems to deliver the proper dosage of drugs or other active ingredients to the designated target spot within the biological cell [20,21]. Similar roles also apply to plant systems where CNMs are used as seed coatings, insecticides, growth promoters, phytohormone transporters, fertilizers, and herbicides [21]. 

Lignin-based compounds are one of the best biodegradable building blocks for carbon nanocarriers. Numerous studies have shown that isolated lignin is a bioactive polymer that is both environmentally friendly and biocompatible [22,23,24,25], which supports the use of lignin as a biological material. As a result, there has been an increase in interest in using lignin’s bioactivities for the synthesis of biological materials [26,27]. The biotechnological uses of lignin have been covered in a number of reviews [28,29,30], including those for delivery methods, bioimaging, tissue engineering, wound healing, antimicrobial agents, coating agents, and others [31]. The nontoxic, antimicrobial properties of lignin were the initial motivators behind its conversion into biomaterials. Since microbial pathogens pose a significant risk to people’s health, lignin’s inherent antimicrobial capabilities make it a desirable biomaterial. Lignin has thus been investigated for a long time as an antibacterial substance. 

There are at least three advantages of using lignin as an antibacterial component. Offering biocompatible and environmentally safe antimicrobials is the first step in addressing the problems in terms of cytotoxicity and antimicrobial resistance of conventional antimicrobial compositions. The second challenge for lignin variolization is thought to be opening new opportunities for high-value applications of concentrated lignin. In terms of minimizing the antimicrobial industry’s carbon footprint, different studies have presented lignin-based nanocarriers (L-NCs) as seed-coating agents, specifically in terms of the biochemical activity of entrapped agrochemicals [32,33,34]. L-NCs are usually thought to be relatively safe [35], but toxicity is still a major concern, and compared to the corresponding animal studies, less is known about how nanoparticles affect seed germination and plant development [36]. Additionally, several researchers have questioned the use of nanoparticles in plants to increase agricultural output due to their unfavorable effects on the environment and living things. Additionally, neither the US Environmental Protection Agency nor the European Food Safety Authority (EFSA) has yet to establish thorough information relating to toxicological and metabolism assessments of doses at their nano level (EPA). To the best of our knowledge, no studies have been published on studying the impact of lignin-based nanocarriers containing carbon nanoparticles (L-CNPs) on plant development and disease control agents when used as a seed treatment.

In this research, we are interested in determining whether L-CNPs might have a protective effect against *F. verticillioides* when applied to maize as seed priming in advance of pathogen inoculation of the stalk. Thus, the primary objectives of this research are to (1) develop L-CNP-based seed treatments and assess their efficiency in preventing *F*. *verticillioides*-mediated stalk rot of maize at laboratory and pot experiments; (2) track alterations in physiological and biochemical responses in treated maize plants and identify potential defense mechanisms against stalk rot; (3) investigate the toxicological effects of L-CNPs at different dosages on male and female ICR mice for 90 days after IV administration.

## 2. Materials and Methods

### 2.1. Reagents

Alkali Lignin was acquired from Sigma-Aldrich (St. Louis, MO, USA), potato dextrose agar (Difco, TX, USA), Hexamethylenetetramine (Fischer Scientific, Pittsburgh, PA, USA), Sodium hypochlorite, and sucrose were also purchased (Difco, TX, USA). The analytical grade of all the chemicals allowed for their use without any further purification, including sodium phosphate buffer (Sigma-Aldrich, St. Louis, MO, USA), Absolute ethanol (Riedel-de Häen, Niedersachsen, Germany), Tris-HCl buffer (Fischer Scientific, Pittsburgh, PA, USA), bovine serum albumin (BSA) (Sigma-Aldrich St. Louis, MO, USA), Formalin solution (neutral buffered, 10%, Merck, Boston, MA, USA), Whatman^®^ qualitative filter paper, Grade 1 (Merck, Boston, MA, USA), and Ridomil Gold SL (2%) (45.3% mefenoxam, SYNGENTA, Wilmington, DE, USA). The Abcam H&E Staining Kit (Hematoxylin and Eosin) (Fischer Scientific, Pittsburgh, PA, USA was used, and the Milli-Q Plus system was used to create deionized water (Millipore, Milford, MA, USA).

### 2.2. Fungal Pathogen and Culture Conditions

Corn plants showing symptoms of stalk rot disease caused by *F. verticillioides* were collected during the 2021 season. *F. verticillioides* was isolated from the infected corn tissues. About 3–5 cm long sections of maize tissues displaying stalk rot disease symptoms were thoroughly washed with distilled water. Maize tissues were then thoroughly cleaned with sterile water after being surface-disinfested in sodium hypochlorite (2%) solution for 3 min. They were then dried on sterile filter paper and plated onto potato dextrose agar (PDA) medium that had been amended with 300 mg/L of streptomycin sulfate. At 26 °C, the plated cultures were incubated for 2 weeks. *F. verticillioides* was isolated and initially identified using a morphological identification key [37]. DNA analysis was used for molecular identification confirmation of the target pathogen following the lab’s previous work [38]. The PCR mixture and thermal conditions were performed according to [27]. In addition, PCR amplification and sequencing of the internal transcribed spacer (ITS) region of rDNA were also performed using the ITS1 and ITS4 primers. PCR reactions were performed in a 25 μL final mixture volume containing 2 μL of 10 ng/μL of genomic DNA, 2.5 μL of 10× PCR buffer, 1 μL of dNTPs 10 mM each, 1.5 μL of 25 mM MgCl2, 0.5 μL each of forward and reverse primers (0.5 mM) and 0.2 μL of Taq DNA polymerase (5 U/μL; Biomatik LLC, Ontario, Canada). The amplification program included an initial denaturalization cycle of 3 min at 94 °C, followed by 35 cycles of 15 s at 95 °C, 30 s at 53 °C, 80 s at 72 °C, and a final extension step of 10 min at 72 °C, in a thermal cycler (Applied Biosystems, Foster, CA, USA). After amplification, the PCR products were analyzed by electrophoresis in a 1% agarose gel. PCR products were purified and sequenced using the same forward and reverse primers. Finally, the experimental sequences of the isolated *F. verticillioides* were deposited in GenBank.

### 2.3. Synthesis and Characterization of Lignin-Loaded Carbon Nanoparticles

A lignin nanoparticle suspension was produced from alkali lignin with hydrochloric acid treatment. In this regard, 4% (*w*/*v*) of alkali lignin in ethylene glycol was maintained under stirring for 2 h at 35 °C following the protocol described by Del Buono et al. [32]. In contrast, the microwave-assisted approach was used to produce carbon nanoparticles (CNPs) [39]. At the end of the experiment, Whatman Filter Paper No. 1 was used to filter the purified CNPs. The CNP-containing filter papers underwent overnight dehydration in an oven. The nanoparticles were scraped out and kept in an airtight container after dehydration. The nanoparticles were later dissolved in deionized distilled water with a pH of 7. 

For the synthesis of lignin-loaded carbon nanoparticles (L-CNPs), 250 mL of a mixture of lignin nanoparticles (2.76 g), hexamethylenetetramine (0.42 g), and 50 mg of carbon nanoparticles was treated with high-power ultrasound for 45 min at 50% amplitude at 90 °C and under argon flow. The precipitated nanoparticles were then centrifuged, rinsed with deionized water three times, and dried after the synthesis. After drying, the produced L-CNPs were ground and dissolved in deionized water with the known working concentrations and left under a stir (1000 rpm) for 1 hr. Using an Ultrasonic processor VCX750 (Sonics & Materials, Inc., Newton, CT, USA), 20 kHz high-power ultrasonic irradiation was carried out with a diameter tip of 13 mm, (Appendix A). In addition, The ultraviolet–visible (UV–Vis) absorption spectra of all the formed nanoparticles lignin, carbon and lignin-loaded carbon nanoparticles samples were recorded on a spectrophotometer (UV-160, Shimadzu Co., Kyoto, Japan). In this regard, the prepared samples were dissolved in a 0.1 M NaOH solution, diluted with deionized water, and the absorbance between 240 and 420 nm was measured. The size distribution and zeta potential analysis were characterized by dynamic light scattering (DLS) with a Zetasizer Ultra (Malvern Panalytical Ltd., Malvern, UK). For TEM analysis. A sample was prepared by diluting one milligram of L-CNPs in one milliliter of distilled water; then, a single drop of the solution was sonicated for 1 h, then placed onto carbon-coated copper TEM grids that had been previously coated and allowed to dry at room temperature for 3 h, while the extra solution was erased using a blotting paper.

### 2.4. Antifungal Activity of L-CNPs

The antifungal activity of the prepared L-CNPs at four different concentrations (25, 50, 100, and 500 mg/L) against *F. verticillioides* was evaluated following the growth rate method to select the more effective concentration of L-CNPs. Briefly, *F. verticillioides* mycelial discs (7 mm in diameter) cultured on potato dextrose agar (PDA) plates were carefully cut from the colony’s plate edges and deposited on the synthetic media plates supplemented with the above concentrations of the formed L-CNPs. The inoculated plates were then incubated for 4, 6, and 8 days at 25 °C ± 1. Following incubation, the mycelial radial growth of *F. verticillioides* was evaluated in all plates, and data were given as the percentage of inhibition [40]:Inhibition rate of fungal growth (%) = [(T − t)/T] × 100(1)
where T represents the *F. verticillioides* mycelial radial growth on the control (untreated) plate, and t represents the *F. verticillioides* mycelial radial growth on the plate treated with different concentrations of L-CNPs. Positive and negative controls included the commercial fungicide (Ridomil Gold SL (2%) and distilled water, respectively. Under carefully monitored conditions, all laboratory experiments were performed in triplicate.

### 2.5. In Vivo Experiments

#### 2.5.1. Inoculum Preparation and Seed Treatment

Tesso et al.’s [41] protocol was modified to produce an inoculum suspension from *F. verticillioides* culture grown in potato dextrose broth (PDB; 0.4% potato starch, 2% dextrose; BD Difco, Sparks, MD, USA). At 30 °C, the spore suspension was incubated and shaken at 160 rpm for 5 days. After shaking, the spore suspension was filtered, and the spore concentration was adjusted to the required dose (1 × 10^6^ conidia mL^−1^) by using a sodium phosphate buffer solution.

#### 2.5.2. Seed Treatment and L-CNPs Effect on Seed Germination Plant Growth

Maize seeds (*Zea mays* L., cv Belgrano) were sterilized for 3 min in a sodium hypochlorite solution (0.25% *w*/*v*). Seeds were then cleaned multiple times with distilled water. To prime maize seedlings, four separate L-CNPs solutions at concentrations of 25 mg L^−1^, 50 mg L^−1^, 100 mg L^−1^, and 500 mg L^−1^ were created. The seeds were subsequently dipped in 10 mL of these solutions for six hours in gentle agitation for seed priming, followed by air-drying prior to sowing with three replicates for each treatment. Treatments with Ridomil Gold SL (mefenoxam, 0.2%) and distilled water were used as seed-soaking solutions. The seeds were then placed on sterilized filter paper in Petri dishes (four seeds per plate), and 10 mL of distilled water supplemented with 1 × 10^6^ conidia mL^−1^ of *F. verticillioides* was added. These samples were covered and placed in a dark (22 ± 2 °C) growing room. The percentage of seed germination and radical lengths were measured after 4 and 5 days post-sowing (DPS), respectively, in Petri dishes.

Continued work was also done for more evaluation of the seed germination and plant growth after 14 days post-sowing after transplanting another similar group of the treated seeds in sterilized plastic pots (10 cm diameter × 20 cm height) filled with a mixture of soil and sand at 3:1 (*w*/*w*), pH 7.0.

### 2.6. Physiological Studies

#### 2.6.1. Chlorophyll, Carotenoids, Anthocyanin, and Soluble Protein Determinations

The physiological effects of L-CNPs on plant height, dry weight, and the level of some photosynthetic pigments (chlorophyll and carotenoid content) were also evaluated in all infected seed maize samples, as described above, and collected after 14 days post-sowing (DPS). For measuring the pigment level, about 100 mg of leaf tissues from each set of treatments, including the mock-treated control, were weighted and extracted with 85% acetone in water (*v*/*v*) until the complete release of chlorophylls and carotenoids into the solution. Then, the suspensions were filtered, and the absorbance was recorded at wavelengths (452.5, 644, and 663 nm). The content of photosynthetic pigments was assessed according to Venkatachalam et al. [42]. To determine the anthocyanin content, 50 mg of the harvested maize shoots were extracted with ethanol (95%) using a pestle and mortar. The resultant suspension was filtered before being centrifuged for 20 min at 8 °C at 7000 rpm. Finally, the anthocyanin concentration was evaluated spectroscopically at 535 and 650 nm, according to Lichtenthaler and Buschmann [43]. The concentrations of carotenoid and chlorophyll (s) were measured as mg/g fresh weight (FW). A cold mortar and pestle were used to homogenize about 50 mg of leaf tissues in 5 mL of 0.1 M Tris-HCl buffer (pH 7.5) for protein assays. At 4 °C, the extract was centrifuged for 15 min at 10,000 rpm. The protein content of the supernatant was then calculated using Bradford’s [44] method using a standard of bovine serum albumin (BSA).

#### 2.6.2. Determination of Acid Invertase (AI) and Protease Activities

The acid invertase activity of maize leaves was measured following the modified approach of Hwang and Heitefuss [45]. After inoculating the pathogen into the control and infected leaf tissues, leaf segments (0.25 g) were taken, submerged in ice-cold ethyl acetate for 20 min, and then rinsed in ice-cold distilled water. Each specimen was incubated for 60 min in a water bath at 30 °C with 0.1 M sodium phosphate buffer (pH 5.6), 0.5 M sucrose, and deionized distilled water. Afterward, each leaf sample was mixed with 2 mL from each 0.5 M sucrose, 0.1 M sodium phosphate buffer (pH 5.6), and 6 mL of double-distilled water after the samples incubation in a shaking water bath at 30 °C for 60 min. AI activity was measured at an absorbance of 280 nm in a spectrophotometer (HITACHI Model: U-1100 573 × 415). On the other hand, protease activity was evaluated following the method of McDonald and Chen [46]. A spectrophotometer was used to evaluate each sample’s protease activity at 660 nm after 30 min. A rise in optical density at 660 nm of 0.1 h^−1^ at 30 °C and pH 7.0 was measured as one unit of protease activity.

#### 2.6.3. Estimation of Peroxidase (POD) and Chitinase Activities

Peroxidase activity for all the treated samples was monitored following the protocol of Vetter et al. [47] and as modified by Gorin and Heidema [48]. Chitinase activity was measured following the protocol of Stangarlin and Pascholati [49]. The supernatant absorbance was measured in each sample [50,51].

### 2.7. Greenhouse Experiments

L-CNPs Activity against Maize Stalk Rot Disease

Following a completely random design, ten treated seeds were then planted in sterile plastic pots (25 cm diameter, 40 cm height) filled with a 3:1 (*w*/*w*) mixture of soil and sand at pH 7.0 under glasshouse conditions. *F. verticillioides*-infected plants were marked with distinctive labels during the tasseling stage (60 days after planting). According to the protocol of Tesso et al. [41], inoculation was carried out by injecting 1 mL of suspension (1 × 10^6^ conidia mL^−1^) into the second node of the growing maize stem using a syringe. The stem was taped off at the inoculation site to create humidity that would promote the fungal pathogen growth. A mock inoculum of 1 mL of sterile phosphate-buffered solution (PBS) was also applied to control plants. All L-CNPs treatments were evaluated in triplicates (six pots in each replicate and five plants per pot, forming a total of 30 plants). All treated plants were examined at the 3–5 leaf stage to confirm whether L-CNPs at different concentrations showed significant activity in the promotion of seedling growth and the inhibition of pathogen infection in the seedling roots. Positive and negative controls for the experiments consisted of seeds treated with the chemical fungicide Mefenoxam (0.2%) and mock-treated with H_2_O, respectively. At 21 days after inoculation, three different plants from each pot were harvested and scored for disease reduction [41].

### 2.8. Toxicity Studies

#### 2.8.1. Sub-Chronic Toxicity Study

All experimental protocols were conducted according to ethical approval (ETHICS446). A total of 40 male and 40 female ICR mice aged five weeks (25–30 g) were used. All animals were held in plastic cages with stainless steel wire-bar lids and kept under standard environmental conditions (23 ± 1 °C, 50–60% relative humidity, and 12 h light/12 h dark cycle) with free access to water and reliable commercial mouse feed. All animals were acclimatized to this environment for at least one week before the sub-chronic toxicity test.

Following an acclimation period of 1 week, animals were randomly divided into two main groups: “Group No. 1 for two weeks with a single injection” and “Group No. 2 for one month with two injections, at zero time and after two weeks”. Each group contained six sub-groups, with four rats in each sub-group. The L-CNPs doses were 50, 100, 200, and 500 mg/L. Approximately 100 μL from each concentration per rat was received for each concentration from the prepared L-CNPs.

Groups of rats were euthanized via CO_2_ asphyxiation followed by thoracotomy, necropsied, and samples collected at 15 and 30 days. The testing of chemicals was conducted following the Safety Assessment Guidelines, which were approved by the Ministry of Health and Welfare in Taiwan [52]. The clinical signs of toxicity and mortality, including convulsion, tremor, vocalization, diarrhea, piloerection, salivation, lacrimation, skin and fur alterations, dyspnea, lethargy, and death, were recorded daily. Throughout the experiment, body mass and food intake were monitored and regularly reported.

#### 2.8.2. Histopathological Examination Studies

For additional histopathological analyses, the liver and kidney tissues of all treatment rates were fixed in 10% neutral formalin. The formalin-fixed tissue was dehydrated in ethanol concentrations of 70%, 80%, 90%, 95%, and 100% before being immersed in paraffin and cut into 5 m thick sections. Hematoxylin and eosin (H&E) were used to stain these sections, which were then inspected under a light microscope (Olympus, Tokyo, Japan).

### 2.9. Statistical Analysis

Statistical analysis was performed using R software program packages (version 4.0.5) to detect significant variations. Each analysis was conducted in triplicate, and the results were presented as means and standard deviations. One-way analysis of variance (ANOVA) was used to test for differences, followed by Duncan’s new multiple range test (*p* < 0.05), and error bars were plotted in the corresponding figures.

## 3. Results and Discussion

### 3.1. Synthesis and Characterization of Lignin-Loaded Carbon Nanoparticles

L-CNPs were produced following the protocols described by Del Buono et al. [32]. The UV spectra of lignin nanoparticles indicated an absorption spectrum around λ = 287 nm, which is quite common for lignin nanoparticles, originating from the aromatic rings/non-conjugated phenolic groups present in the lignin structures. While an absorption peak appeared at λ 235 nm in the case of the carbon nanoparticles, a clear UV shift was produced at absorption spectrum λ 250 nm after loading lignin with carbon nanoparticles, which thus indicated the loading of carbon on the lignin nanoparticles.

Data from TEM analysis indicated that the produced L-CNPs possessed a regularly spherical structure with a relatively smooth surface (Figure 1A). The DLS analysis showed that L-CNPs have 52 ± 1.2 nm size and a −43.5 ± 0.5 mV zeta value (Figure 1B). Furthermore, the zeta potential absolute value indicates a high electrical charge on the L-CNPs’ surface, causing a strong repulsive force between the particles to avoid agglomeration and, therefore, might be responsible for their elevated stability. On the other hand, the L-CNPs’ components were produced with 39.4 ± 2.1 nm size and −40.5 ± 1.3 mV zeta value for lignin nanocarriers (L-NCs) and 15.3 ± 4.5 nm size and −40.5 ± 0.42 mV zeta value for carbon nanoparticles (C-NPs) (Appendix A), indicating an ideal surface charge. Carbon NPs were immobilized individually on the surface of Lignin NCs with the aid of a high-power ultra-sonication process and the presence of a hexamethylenetetramine as a surfactant. The L-CNPs nanoparticles can be observed as thin semitransparent sheets decorated with many dark-colored sports, likely C-NPs, which may be firmly immobilized on the surface of the L-NPs nanosheets (Figure 1A).

### 3.2. Molecular Identification of the Fungal Pathogen Fusarium verticillioides

*F. verticillioides* genomic PCR analysis was confirmed using the pair of primers “uni-F/uni-R” with the given DNA products with sizes around 723 bp, which confirmed their identity as *F. verticillioides* strains. When the ITS region was amplified using both primers ITS1/ITS4, 430 bp to 540 bp of DNA were produced. The genomic sequence of the *F. verticillioides* amplification product showed e-values of 0 and 100% similarity with the GenBank *F. verticillioides* accessions no. KF031434.1, which confirmed the morphological characterization. The partial sequences of the ITS regions from *F. verticillioides* isolate were deposited in the GenBank with an accession no. OP870085.1 (Appendix A).

### 3.3. Activity of L-CNPs against F. verticillioides

Next, the antifungal activity of the prepared L-CNPs was tested at four different concentrations (25, 50, 100, and 500 mg/L), which were subsequently mentioned as L-CNPs25, L-CNPs50, L-CNPs100, and L-CNPs500, respectively, against the mycelial growth rate of *F. verticillioides* in vitro (Figure 2A). The percentage of inhibition in the mycelial fungal growth rate for *F. verticillioides* compared to the treatment with chemical fungicide (Ridomil Gold SL, 2%) on 4, 6, and 8 days was also demonstrated in Figure 2B. Significant inhibition percentages of 91.23 ± 1.35, 95.23 ± 1.35, 96.22 ± 2.10, and 98 ± 1.25% (Plates 5–8) were acquired on the 8th day of incubation at L-CNPs concentrations of 25, 50, 100, and 500 mg/L, respectively, compared to inhibition rates of 0.0, 77.12 ± 0.3, 55.19 ± 2.5 and 50.89 ± 2.5 (Plates 1–4) in the cases of the control, the chemical fungicide, lignin nanoparticles alone, and carbon nanoparticles alone, respectively.

### 3.4. Effects of Lignin Nanoparticles on Seed Germination and Maize Growth

After 4 days, the effects of L-CNPs at 25, 50, 100, and 500 mg/L concentrations on the germination percentage of the treated seeds were evaluated and compared to the chemical fungicide treatment and both controls (untreated non-infected seeds and untreated infected seeds) (Table 1 and Figure 3A). Based on the data presented in Table 1, the L-CNPs25, L-CNP50, and L-CNPs100 considerably had an inductive impact on seed germination and the radical length compared to the chemical fungicide treatment. L-CNPs100 had the most positive effect on this metric compared to L-CNPs25 and L-CNP50 (Table 1). At the same time, the maximum dose (L-CNPs500) had a suppressive effect on seed development. Comparable results were also recorded for the seed germination after 15 days of sowing in another group of treated seeds and transplanting them in sterilized plastic pots filled with a mixture of soil and sand at 3:1 (*w*/*w*). This indicates that L-CNPs100 concentration is the better concentration in increment seed germination (See Table 1 and Figure 3B). This may be backed to that lignin extracts derived from Giant Reed have already been demonstrated to have beneficial impacts on seed germination [53]. In addition, the noticeable radicle formation increment is thought to be hormonal and caused by lignin structural characteristics [53]. Lignin humic-like action, for example, can have a significant impact on germination and radicle length [54].

The content of the L-CNPs’ matrix has been used to illustrate the stimulatory activity. Phenol-rich compounds, like lignin, have been shown to boost metabolic reactions, especially when decorated with inductive particles like CNPs, resulting in enhanced seed growth [53]. This impact can be ascribed to lignin and its phenolic units, which have a hormone-like influence on the metabolic processes of the seed developing [55]. Likewise, lignin-based molecules can also have a gibberellin-like impact, altering the hormonal condition of the seed and the biological functions that underpin its growth [53].

Furthermore, because of its ability to operate directly on the mitotic index, lignin has been proposed to promote seed germination [56]. Falsini et al. [57] proposed that the numerous hydrophilic groups in the lignin chemical composition had positive impacts on the growth of different plants. Contextually, these groups, which are found on the outer nanoparticles’ surface, can potentially improve plant germination and development by improving water availability. In this regard, it was suggested that the produced L-CNPs may reduce the frequency of aromatic and phenolic rings, meaning that the aromatic rings were oxidized to quinones rings. Furthermore, throughout the acid treatment operation, the number of peaks ascribed to the aldehyde decreased, indicating that some ester bonds or carboxyl groups were generated [58]. When pristine lignin was acid-hydrolyzed, the average molar mass decreased, resulting in a higher poly-dispersity index than the pristine micro-lignin, which indicates a greater presence of smaller molecular weight particles and consequently improving plant growth [58]. 

On the other hand, other reports have found that lignin-based molecules have little influence on seedling growth [20,59]. As a result, the type, content, formulation, and concentrations of lignin have affected this critical stage of plant growth. Considering their internal structures and tiny size (52 ± 1.2 nm) [58], these findings suggest that L-CNPs at concentrations of 25, 50, and 100 mg/L may have been more effectively absorbed by the maize seeds than other raw lignin precursors. Therefore, L-CNPs play a substantial biological role in maize seed germination and plant growth. However, the excessive L-CNPs amount, in the case of L-CNPs500, was harmful and hindered seed growth. The increased concentration of lignin-phenolics caused this negative effect. Similarly, several substances have been shown to influence the early phases of plant growth, as evidenced by the seeds’ sensitivity to high levels of polyphenols [60]. Finally, efficient seed priming can improve the ability of immature plantlets to adapt to external cues and stress [61].

### 3.5. Effects of L-CNP-Treated Maize Seeds on F. verticilloides-Infected Plants

In the pot experiment, it was noticed that seed treatment with L-CNPs can impact the plant’s subsequent development and effectively inhibit/reduce the infection with *F. verticillioides* to control stalk rot disease infecting maize plants compared to treatment with the recommended chemical fungicide (Figure 4). A significant disease reduction in maize stalk rot ranged from 33 to 86% for all L-CNPs concentrations compared to the infected control (Figure 4). Treatments with the two highest L-CNPs concentrations (100 and 500 mg/L) showed a significant disease reduction by 86% and 81%, respectively, compared to 79% for treatments with the chemical fungicide. However, treatment with either lignin nanoparticles or carbon nanoparticles alone showed lesser disease resistance at only 20 and 40%, respectively. This was also reported by Figueiredo et al. [62] and Campobenedetto et al. [63]. Consequently, these favorable effects on earlier growth and maturation are important.

### 3.6. Physiological Parameters

#### 3.6.1. Effect on Root and Shoot Length

Two weeks after seeding, the maize seedlings were measured for root and shoot lengths and weights (Figure 5A–D). The lowest average L-CNPs dosages, L-CNPs25, L-CNPs50, and L-CNPs100, had a beneficial influence on the maize’s ability to produce biomass overall (Figure 5A–D). Other treatments, however, were ineffectual or even detrimental in this regard. Plant biomass improved due to the lignin and phenolic compound treatment [64,65]. Syringyl phenols, which have gibberellin-like action, can stimulate seedling growth [66].

The nanoparticles used in the current work have a high amount of syringyl groups [58]. As a result, the improvements in biomass production are consistent with the findings of Nardi et al. [66] and Savy et al. [53], who discovered that lignin substitutes enhanced maize growth but did not affect germination. The existence of guaiacyl and p-hydroxy-phenyl groups in lignin can positively impact plant growth [53]. These two components can help with root system growth and biomass [67]. Furthermore, materials comprising phenolics promote plant development by enhancing the seedling intake of the key macro- and micro-nutrients. In this regard, Ertani et al. [64] demonstrated that phenols increased the absorption level of maize from N, P, Ca, K, and Mg by activating the root system.

L-CNPs at different concentrations (25, 50, 100, and 500) mg/L differently affected shoot and root length in maize (Figure 5A,B). In terms of both shoot and root length, it was clearly noticed that the applied L-CNPs treatments, particularly L-CNPs25, L-CNPs50, and L-CNPs100, were able to stimulate the shoot length. However, L-CNPs500 and the chemical fungicide treatments showed ineffective activity in both the shoot and root length. The different treatment solutions were then measured on the maize’s shoot and root fresh weight (Figure 5C,D). The obtained data were also in parallel with the root and shoot length data, where all LNP treatments did not passively affect root and shoot fresh weight. The only exception was that the L-CNPs500 treatment decreased the root and shoot fresh weight parameters.

#### 3.6.2. Chlorophyll, Carotenoid, Anthocyanin, and Soluble Protein Content Shown by Maize Treated with Lignin Nanoparticles

The obtained results indicated that L-CNP-treated maize seeds have significantly increased chlorophyll a and b content. Different effects were seen in maize samples treated with various L-CNPs concentrations in terms of chlorophyll a and b content (Figure 6A,B). In particular, the L-CNPs100 concentration induces a >50% increment in chlorophyll and around 40% in chlorophyll b compared to the control. L-CNPs500 also derives an increase in chlorophyll a and b, but with comparatively much smaller percentages. Concerning chlorophyll b, all treatments increased the content of this pigment (Figure 6B). In particular, seeds treated with either L-CNPs25, L-CNPs50, or L-CNPs100 provoked the highest increases if compared to those treated with the chemical fungicide or untreated control samples.

All of the treatments resulted in significant increases in maize chlorophyll content. L-CNPs’ inductive effects on chlorophyll a and b are due to a stimulatory influence on their production. Due to their ability to promote plant nutrition, particularly nitrogen absorption, several phenolic compounds can increase the concentration of the pigments described above [68]. Furthermore, strong root development, as urged by L-CNPs25, L-CNPs50, and L-CNPs100, might justify the maximum chlorophyll a and b concentration since a superior root system allows maize to enhance its ability to take up nutrition from the growth media [69]. Finally, greater chlorophyll concentrations are thought to be a natural physiological response to environmental stimuli, enhancing the crop’s ability to capture light in photosystems [70].

The chlorophyll a/b ratio has been proposed as a metric for assessing crop response to environmental adversities, diseases, and stressors [71]. Some stresses, in particular, can cause the conversion of chlorophyll a to b via the enzyme Chl an oxygenase [72]. As a result, the increase in chlorophyll b, which was not offset by a sufficient rise in chlorophyll a, demonstrates that the higher L-CNPs dose had phytotoxic effects. Figure 6C depicts the results obtained in terms of carotenoids. L-CNPs25, L-CNPs50, and L-CNPs100 treatments resulted in significant increases in carotenoids, but the L-CNPs500 doses had little effect on the carotenoid levels. In the case of anthocyanin (Figure 6D), the L-CNPs25, L-CNPs50, and L-CNPs100 treatments significantly increased its content. L-CNPs500 and the samples treated with the chemical fungicide also increased the anthocyanin content compared to control samples, although the magnitude of these increases was much smaller than that induced by the first three concentrations mentioned above.

Figure 6C,D compares the results of the samples treated with different dosages of L-CNPs to the controls in terms of carotenoid and anthocyanin. Carotenoids are light-harvesting pigments that play a role in photosynthesis and have antioxidant properties. As a result, they serve a critical function in helping plants eliminate reactive oxygen species (ROS) [73]. Carotenoids, in particular, protect chloroplasts from ROS by quenching chlorophyll in singlet and triplet forms [74]. L-CNPs at (25, 50, and 100 mg/L) concentrations have a stimulatory effect on carotenoid production, according to the obtained findings. The recognized rise in carotenoids can be attributed to a protective response to LNP therapy. The concentration of chlorophyll is also correlated to the number of carotenoids involved. Increases in the level of this pigment are perceived by plants as a signal, which drives carotenoid production [74]. Based on these findings, using non-excessive dosages of LNP as a way to increase the synthesis of these beneficial compounds is recommended. In terms of anthocyanin, increases were seen in maize following various LNP treatments. Anthocyanin is a flavonoid that has essential functions in plants as well as potential health benefits for humans [75]. Anthocyanin can prevent and contrast lipid peroxidation by acting on ROS in the vacuole, and their content can be elevated in response to diverse environmental conditions [76].

Regarding these findings, maize interpreted L-CNPs at 25, 50, and 100 mg/L as a non-phytotoxic signal, which increased the anthocyanin concentration. Anthocyanins are formed in the biosynthetic pathways [75], and their increased levels after exposure to LNP can be ascribed to the ability of substances containing phenolic compounds to activate the metabolic route mentioned above [64]. Furthermore, an increase in chlorophyll content has been connected to an increase in anthocyanin, suggesting that an increase in chlorophyll content is a potential physiological response to compensate for anthocyanin increases [70]. The minimal rise in anthocyanin at the maximum doses of LNP was due to the treatment’s phytotoxic impacts, which limited the plant’s capacity to respond to the stress.

#### 3.6.3. Determination of Soluble Protein

In terms of soluble protein, it was noticed that L-CNPs25, L-CNPs50, and L-CNP100s had the greatest rise in their content (Figure 7). Other assays had no significant effect on soluble protein compared to the control samples. The final biochemical variable assessed in response to L-CNPs treatments was the total soluble protein content, as this parameter is highly correlated with the plant’s potential to absorb nutrients and can also be utilized to determine phytotoxicity [42,77]. Only L-CNPs at 25, 50, and 100 mg/L concentrations can considerably enhance the soluble protein content compared to the control. The favorable impact of phenols on maize seedlings might explain this good effect. As indicated previously, they can improve the nitrogen content of treated plants, which is essential for protein biosynthesis [64]. The noticeable decrease of soluble protein in L-CNPs500 might result from the phytotoxic effects induced by this treatment. In reality, this dose is thought to have disrupted protein biosynthesis or caused oxidative stress, which led to protein degradation [42]. Due to their capacity to penetrate mitochondria or other cellular organelles and interfere with crucial cellular processes, large amounts of lignin nanoparticles have also been proven to have negative consequences [78].

### 3.7. Influences in Defense Enzyme Activity of Maize Leaf after L-CNP Seed Treatment

Various defense-related enzymes were examined to understand more about the interaction of *F. verticillioides* with maize following the topical applications of L-CNPs. Data obtained showed that the inoculation of *F. verticillioides* improved the activity of all the defense-related enzymes examined in all L-CNPs treatments compared to the uninoculated control plants. In the pot experiment, the inoculated plants exhibited higher leaf acid invertase (AI) and protease activity than the untreated plants. However, it was noticed that the gradual increase of L-CNPs concentrations of seed treatments significantly increased the acid invertase and protease activity compared with the infected control. The maximum increase in AI and protease activity was given by L-CNPs at 100 mg/L (Figure 8A,B).

In contrast, *F. verticillioides* inoculation stimulated POD and chitinase activity in seed-treated plants more than in untreated controls (Figure 8C,D). Compared to the infected control in the pot, the L-CNPs seed treatment significantly increased the leaf POD and chitinase activities (Figure 8C,D). Thus, the L-CNPs treatment had a stimulatory effect on these enzyme activities over the pathogen effect. The maximum increase for both POD and chitinase activity was given by L-CNPs100, followed by L-CNPs500. At the same time, Ridomil Gold SL (2%) showed an increase in both POD and chitinase activity.

These findings suggest that maize is more resistant to *F. verticillioides* infections when L-CNPs are applied exogenously. This is in line with other reports showing that some derived compounds applied exogenously can induce resistance in their hosts by increasing higher levels of host defense enzymes and pathogen-related (PR) proteins. This suggests that the L-CNPs induce systemic resistance in the host plants, which makes them less likely to get infected [79].

Systemic acquired resistance (SAR), a plant immune response that prevents infection or disease from extending to the host’s non-infected parts, is typically made up of PR proteins like chitinases, -1,3-glucanases, acid invertases, and peroxidases because the onset of induced systemic resistance in plants correlates with increased activity and expression of these proteins [80,81,82]. Many crops can develop more plants and experience a reduction in disease thanks to substances obtained from plants, according to numerous research [83,84]. Plant defense genes that are dormant in healthy, uninoculated plants can result in systemic resistance to disease when they are awakened by a variety of conditions.

According to numerous reports, increasing levels of chitinase, protease, POD, AI enzyme activities, and the expression of the genes for -1,3-glucanase and chitinase are effective against a variety of fungal diseases [72,83,84,85]. In this regard, Lanubile et al. [86] published a thorough assessment of maize genes involved in pathogen detection, the defense signaling network, and the functions of enzymes that support host (maize) resistance to *F. verticillioides*. By increasing the levels of defense-related enzymes and pathogenesis-related proteins, L-CNPs seed treatments at various concentrations provided defense in maize plants against the stalk rot pathogen. These proteins may be crucial in fortifying the host plant’s cell walls to resist *F. verticillioides* infection. It is also proposed that L-CNPs may activate their defense mechanisms in response to pathogen inoculation by developing additional proteins to prevent the entry of the pathogen or its subsequent spread because L-CNPs consist of carbon and lignin, which is generated by plants [87].

### 3.8. Toxicity Evaluations

#### 3.8.1. Mortality and Clinical Observations

During the 90-day observation period and throughout the L-CNP treatments, neither sex of mice experienced any deaths or unusual clinical manifestations or behavioral alterations, such as skin and hair problems, alterations to the eyes, diarrhea, stomach breathing, lethargy, or tumors. The normal behaviors and relative body weight increases in the male and female mice among the groups (See Table 2) suggested that L-CNPs had no or low toxicity to mice after intravenous injection [88]. This was in line with clinical data indicating that only a small percentage of treated patients (more than 500,000) displayed extremely brief hyperpyrexia following localized nanoparticle injection [89]. The histopathology examinations were then applied to confirm the low toxicity of the L-CNPs.

#### 3.8.2. Toxicity and Histopathological Studies

Table 1 displays the respective liver and kidney weights of male and female mice. Despite the exceptionally elevated kidney weight in female mice by injection of the generated L-CNPs at 25 mg/L, no significant variations were noticed in the mean body weights of either male or female mice in any of the L-CNP-treated groups in comparison to the control mice. Once compared to mice in the placebo group, there were no morphological changes in the liver or kidney characteristics of either male or female mice treated with L-CNPs at any concentration, as shown in Figure 9A,B. Further evidence that lesions had no negative consequences on either male or female mice after 90 days of therapy came from the lack of glaring lesions of many other essential organs in either of the L-CNP-treated groups versus the control group during necropsy. Most importantly, upon exposure to different doses of L-CNPs for 30 and 90 days, neither male nor female mice showed any functional alterations, which was compatible with histological findings (Figure 9), suggesting that no biological damage was caused by L-CNPs therapies after intravenous administration.

No apparent histopathological alteration was observed for the liver (Figure 9, upper) and kidney (Figure 9, lower) tissues upon the HE staining under the optical microscope. While at L-CNPs500, a slight infiltration of inflammatory cells occurred in male mice administered in both liver and kidney tissues at the first 30 days while disappeared after 90 days (Figure 9, upper and lower arrowed). The inflammatory disappearance may be attributed to the high dispersibility of L-CNPs and their small particle sizes in contrast to other studies [90,91,92], which indicated a significant observed high inflammatory cell infiltration in organ sections treated with carbon nanotubes (CNTs) at high doses and larger particle sizes.

On the other hand, other carbon nanomaterials, such as GO, and other graphene quantum dots, did not induce any toxicity or oxidative stress after intravenous exposure [93]. In this respect, even though carbon is regarded as inert for biological systems, the potential toxicity of nanoparticles for human and animal health is still a major issue. For further application in food and medicine, it is required to determine any possible long-term harmful consequences from either oral or intravenous administration.

For instance, the body uptake and tissue distribution of some metallic nanoparticles, such as gold nanoparticles (AuNPs) in the proper form, size, and stabilizing agents, may be influenced. The maximum uptake was seen for AuNPs with a diameter between 25 and 50 nm, while particles larger than 50 nm exhibited decreased absorption [18]. Together with the obtained results, the available data indicated that L-CNPs did not cause any sign of toxicity, particularly at 25, 50, and 100 mg/L doses. The properties of carbon nanomaterials, particularly type, size, shape, and concentration, and their regulation rules require further investigation.

## 4. Conclusions and Future Perspectives

Lignin is one of the three principal components of plant cell walls, accounting for 10–40% of lignocellulosic biomass dry mass. It is typically emitted as waste by biomass refinery businesses. Recently, there has been a rise in interest in exploiting lignin’s bioactivities for the manufacture of biological materials. Several reviews have examined the biomedical and biotechnological applications of lignin, such as delivery vehicles for insecticides and other biomolecules, bio-imaging, tissue engineering, coating agent, wound healing, and others, demonstrating the substantial value of these molecules in various domains. 

In this paper, and for the first time, the impact of nanostructured lignin microparticles on a crop was assessed. The results indicated that L-NPs may have potential use in agricultural applications as seed treatments serving as suitable delivery systems for many bioactive compounds, including carbon nanoparticles or any other fungicides. L-CNPs-primed maize seedlings elicited intriguing physiochemical responses with little to no effect on seed viability and plant growth. The results imply that nanostructured lignin microparticles at specified dosages (L-CNPs25, L-CNPs50, and L-CNPs100) might be a potential technique for stimulating favorable biological responses in plants and also induce disease resistance. However, higher doses (L-CNPs500) reduced efficiency or even caused potentially toxic consequences. Too high levels of lignin component units likely outweighed the positive effects as the L-NP concentration grew (p-hydroxyphenyl, guaiacyl, and syringyl). Excessive levels of these elements have been shown in the literature to hinder plant growth [94]. Regarding the possibility of this biomass, further study is needed to fully comprehend the nature of both beneficial and adverse consequences. In conclusion, lignin-polymer-based nanoparticles can lead toward a safe and natural alternative to improve pesticide use in seed treatment without interfering with plant physiology. This is especially important because of the wide availability of this biomass, the potentially negative effects of its management on the environment, and the urgent need to create new bio-stimulating substances for agricultural use.

## Figures and Tables

**Figure 1 polymers-15-01193-f001:**
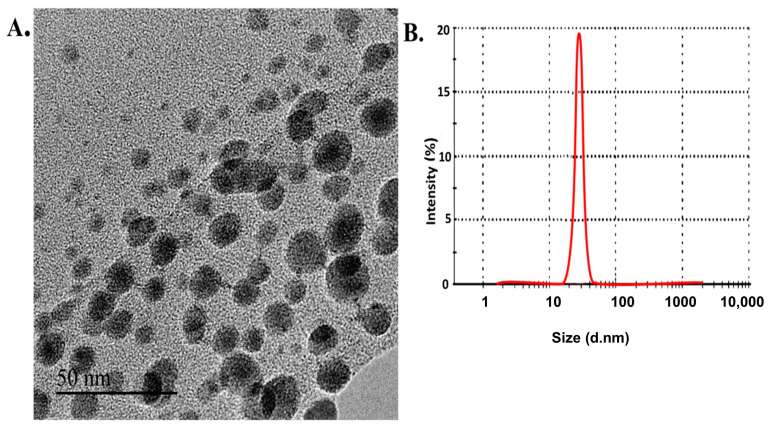
Characterization of lignin-loaded carbon nanoparticles (L-CNPs): (**A**) transmission electron microscope image for lignin nanoparticles (L-NPs) decorated with carbon nanoparticles (C-NPs) producing L-CNPs; (**B**) DLS analysis of L-CNPs.

**Figure 2 polymers-15-01193-f002:**
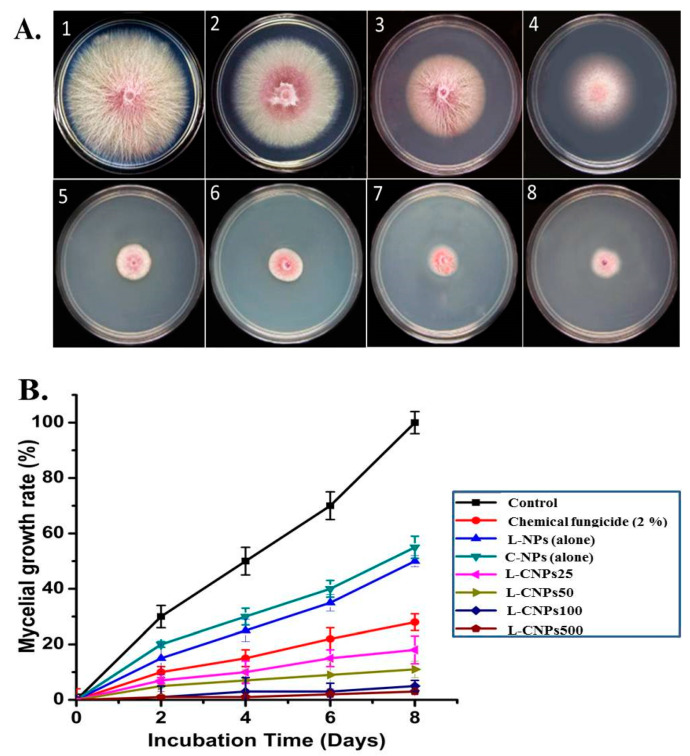
Antifungal activity of L-CNPs against *F. verticillioides*: (**A**) Mycelial growth inhibition of *F. verticillioides* on PDA plates after 8 days of incubation: (1) control, (2–4) carbon nanoparticles alone, Lignin nanoparticle alone, and chemical fungicide respectively; (5–8) carbon nanoparticles alone respectively. (**B**) Mycelial inhibition of L-NPs (alone), C-NPs (alone), chemical fungicide (2%), L-CNPs at different concentrations (25, 50, 100, and 500) mg/L and mentioned by L-CNPs25, L-CNPs50, L-CNPs100, and L-CNPs500, respectively.

**Figure 3 polymers-15-01193-f003:**
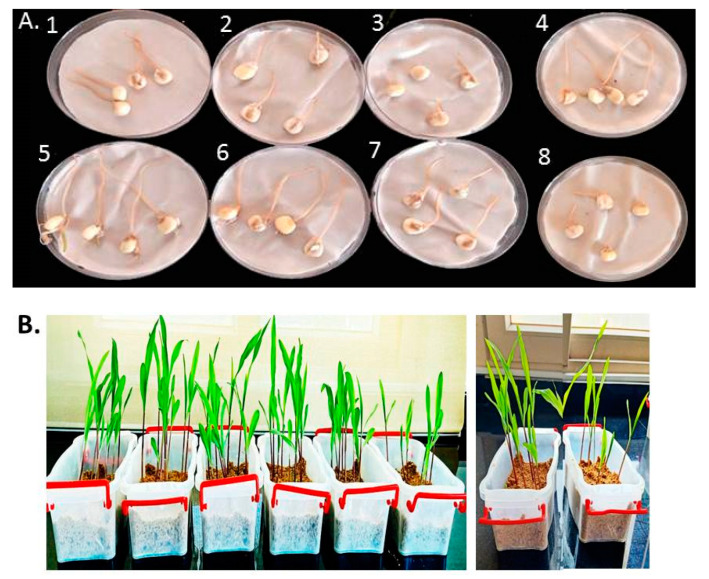
Representative seeds 4 days after treatment with the various L-CNPs concentrations (25, 50, 100, and 500) and refer to L-CNPs concentrations applied to seed for priming (**A**); representative samples 14 days after the treatment with L-CNPs (**B**).

**Figure 4 polymers-15-01193-f004:**
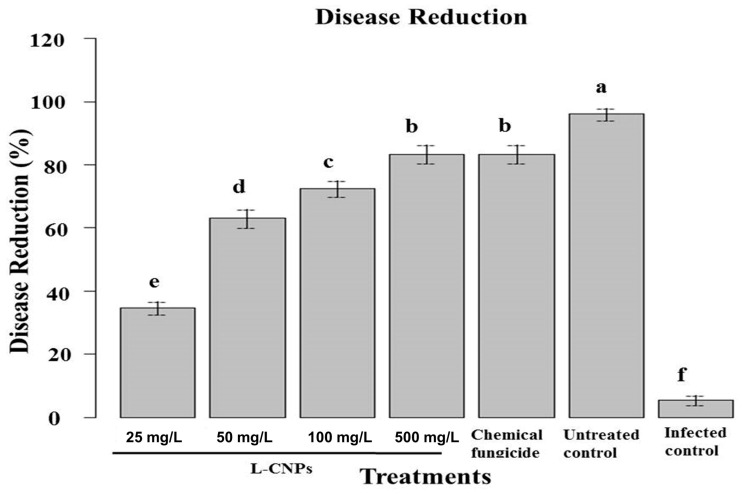
Effect of seed treatments with L-CNPs at different concentrations on disease reduction in maize inoculated with *F. verticillioides* (21 days after inoculation) in pot experiments. Data are expressed as mean  ±  standard deviation (SD) of three replicates. Mean values with different letters are significantly different according to the least significant difference (LSD) test at *p*  <  0.05.

**Figure 5 polymers-15-01193-f005:**
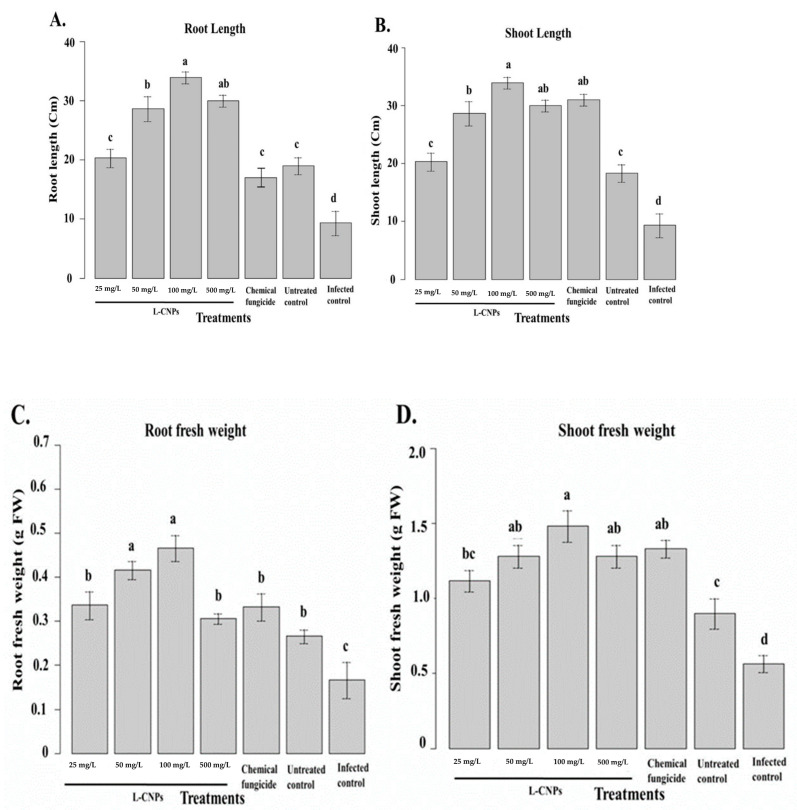
Effect of the treatment with L-CNPs on the length and fresh weight of maize samples compared to the chemical fungicide and the untreated controls. (L-CNPs25, L-CNPs50, L-CNPs100, and L-CNPs500) refer to L-CNPs concentration applied to seed for priming (**A**); effects of the treatments with L-CNPs on the shoot length; (**B**) shoot length; (**C**) root fresh weight; and (**D**) shoot fresh weight compared to the untreated control. The values were recorded on seedlings at 14 DAS. Letters in the figure, if different, indicate statistically significant differences of *p* < 0.05 between treatments.

**Figure 6 polymers-15-01193-f006:**
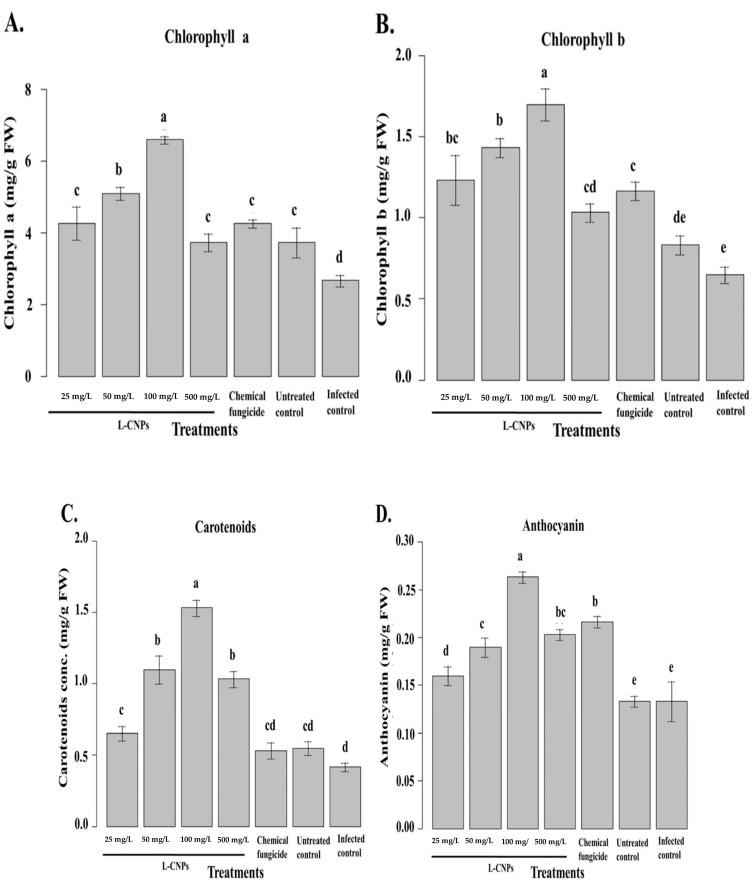
Effect of the treatment with L-CNPs on the photosynthetic pigments of maize samples compared to the chemical fungicide and the untreated controls. (L-CNPs25, L-CNPs50, L-CNPs100, and L-CNPs500) refer to L-CNPs concentration applied to seed for priming (**A**); effects of the treatments with L-CNPs on the chlorophyll a; (**B**) chlorophyll b; (**C**) carotenoids, and (**D**) anthocyanin compared to the untreated control. The values were recorded on seedlings at 14 DAS. Letters in the figure, if different, indicate statistically significant differences of *p* < 0.05 between treatments.

**Figure 7 polymers-15-01193-f007:**
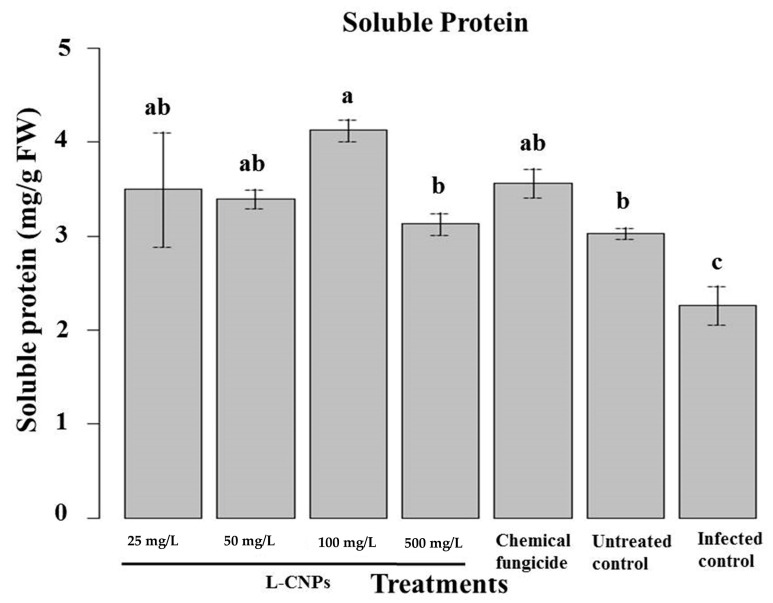
Effect of the treatment with L-CNPs on the available soluble protein of maize samples compared to the chemical fungicide and the untreated controls. (L-CNPs25, L-CNPs50, L-CNPs100, and L-CNPs500) refer to L-CNPs concentrations applied to the seed for priming. The values were recorded on seedlings at 14 DAS. Letters in the figure, if different, indicate statistically significant differences of *p* < 0.05 between treatments.

**Figure 8 polymers-15-01193-f008:**
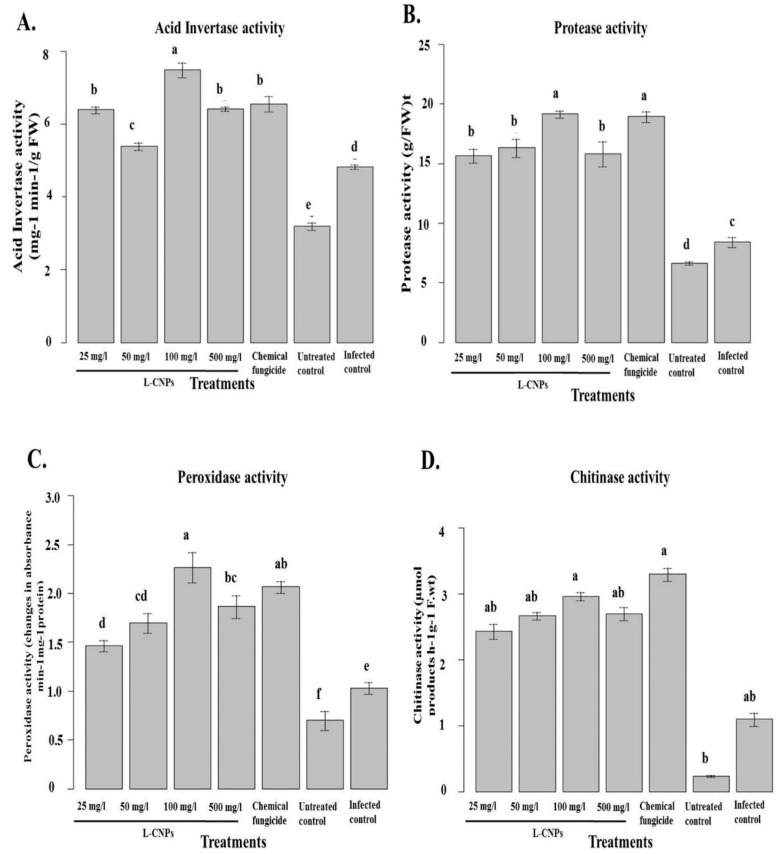
Effect of the treatment with L-CNPs on the defense-related enzymes in maize leave samples compared to the chemical fungicide and the untreated controls. (L-CNPs25, L-CNPs50, L-CNPs100, and L-CNPs500) refer to L-CNPs concentrations applied to the seed for priming. The values were recorded on seedlings at 14 DAS. (**A**) Acid invertase activity; (**B**) protease activity; (**C**) peroxidase activity, and (**D**) chitinase activity compared to the untreated control. Letters in the figure, if different, indicate statistically significant differences of *p* < 0.05 between treatments.

**Figure 9 polymers-15-01193-f009:**
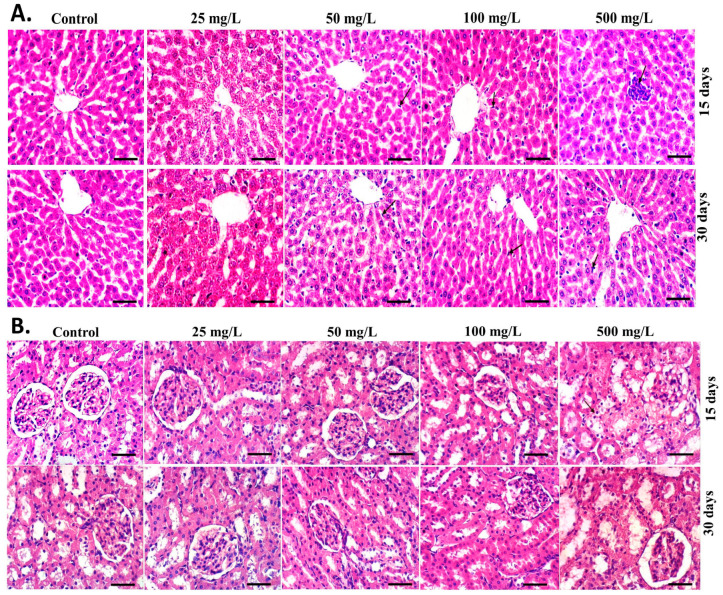
Effects of IV administration of L-CNPs at different concentrations (25, 50, 100, and 500) mg/L for 30 and 90 days on liver and kidney histology in male and female mice. (**A**) H&E staining of (**A**) liver tissues, (**B**) kidney tissues. Scale bar = 50 μm.

**Table 1 polymers-15-01193-t001:** Effects of the treatments with different concentrations of L-CNPs on maize seed germination and radicle length (L-CNPs25, L-CNPs50, L-CNPs100, and L-CNPs500 and applied to seed for priming). The germination was recorded at 4 and 15 days after the treatments, while the radicle length was recorded at 5 days.

Treatments	Germination (%) 4 DPS	Radical Length (cm)	Germination (%) 15 DPS
L-CNPs25	89.0 ^a^	3.16 ^c^	89.0 ^a^
L-CNPs50	94.7 ^ab^	3.27 ^c^	94.0 ^ab^
L-CNPs100	98.6 ^b^	3.27 ^c^	98.6 ^b^
L-CNPs500	65.0 ^c^	1.0 ^a^	52.0 ^c^
Control	87.0 ^a^	2.5 ^b^	86.0 ^a^
Infected	55.0 ^c^	1.0 ^a^	57.0 ^c^
Chemical fungicide (Ridomil plus SL 2%)	84.0 ^a^	1.75 ^ab^	84.0 ^a^
Carbon nanoparticles “alone”	85 ^a^	2.2 ^b^	85 ^a^
Lignin nanoparticles “alone”	82 ^a^	2.0 ^b^	83 ^a^

Mean values within and between treatments with different labels were significantly different (*p* < 0.05) according to one-way ANOVA followed by Tukey’s post-hoc test.

**Table 2 polymers-15-01193-t002:** Body weight gain and relative organ weight of male and female ICR mice 90-day injection administration of L-CNPs.

	Gender	Control	L-CNPs Concentrations			
			25 mg/L	50 mg/L	100 mg/L	500 mg/L
Body weight						
Initial body Weight (g)	Male	32.4 ± 1.2 ^a^	34.0 ± 1.3 ^a^	32.8 ± 1.9 ^a^	34.1 ± 1.3 ^a^	34.1 ± 1.0 ^a^
	Female	15.8 ± 0.5 ^a^	27.3 ± 1.0 ^a^	27.3 ± 2.2 ^a^	27.2 ± 1.0 ^a^	26.5 ± 3.0 ^a^
Final body weight (g)	Male	45.2 ± 1.3 ^a^	44.2 ± 4.1 ^a^	41.1 ± 3.0 ^a^	42.0 ± 4.0 ^a^	42.2 ± 4.0 ^a^
	Female	34.2 ± 0.6 ^a^	34.8 ± 2.8 ^a^	35.7 ± 3.6 ^a^	33.4 ± 2.5 ^a^	33.0 ± 1.3 ^a^
Weight gain	Male	12.8 ± 1.3 ^a^	10.2 ± 1.3 ^a^	8.3 ± 1.0 ^a^	7.9 ± 1.3 ^a^	8.1 ± 2.5 ^a^
	Female	18.4 ± 1.2 ^a^	7.5 ± 2.5 ^a^	8.4 ± 2.5 ^a^	6.2 ± 0.5 ^a^	6.5 ± 1.3 ^a^
Relative organ weight						
Liver (%)	Male	5.3 ± 0.4 ^a^	4.96 ± 0.37 ^a^	5.09 ± 0.24 ^a^	5.10 ± 1.04 ^a^	4.98 ± 2.5 ^a^
	Female	4.5 ± 0.6 ^a^	4.98 ± 0.57 ^a^	4.46 ± 0.45 ^a^	4.79 ± 0.24 ^a^	4.81 ± 0.34 ^a^
Kidney (%)	Male	1.5 ± 0.3 ^ab^	1.81 ± 0.34 ^ab^	1.79 ± 0.18 ^ab^	1.67 ± 0.14 ^ab^	1.72 ± 0.14 ^b^
	Female	1.2 ± 0.1 ^b^	1.52 ± 0.20 ^ab^	1.44 ± 0.18 ^b^	1.48 ± 0.14 ^b^	1.45 ± 0.2 ^b^

Mean values within each column with different labels were significantly different (*p* < 0.05) according to one-way ANOVA followed by Tukey’s post-hoc test. The relative organ weight was expressed as a percentage of body weight.

## Data Availability

Not applicable.

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
