# Peer review of "Lignin-Loaded Carbon Nanoparticles as a Promising Control Agent against Fusarium verticillioides in Maize: Physiological and Biochemical Analyses"

_polymers, 2023, doi:10.3390/polym15051193_

Round 1
Reviewer 1 Report
I suggest rethink the title; "lignin-decorated" sounds unusual.
I suggest a language revision, especially for the Abstract section.
The statement "A naturally occurring biopolymer called lignin..." should be change to "The natural occurring biopolymer lignin...".
Reviewer 2 Report
The authors have done a good job in presenting the results on lignin-based functional materials. The logic is okay, and the presentation is acceptable. However, the writing needs to be significantly improved in terms of language usage, clarity enhancement, etc. For example, the abstract may need to be improved in terms of conciseness. Also, tthe current title is much too broad. Lignin-related papers published in the Journal of Bioresources and Bioproducts, Paper and Biomaterials, and other related journals, in the past 3 years, may be cited to highlight the idea of the current work. Please proof-read the manuscript thorougly to improve the quality of the current work.
Reviewer 3 Report
The manuscript titled “Lignin-Decorated Carbon Nanoparticles: A Promising Tool to Induce Defense-Related Enzymes and Disease Resistance in Maize against Fusarium verticillioides” by El-Ganainy, S.M.; et al. is an original scientific work where the authors characterize lignin-load carbon nanoparticles (L-CNPs) and further assess the capabilities of these nanoparticles by multitude of complementary techniques as scanning electron microscopy (SEM), Z-potential measurements, enzymatic activity assays, histopathological examinations and PCR genomic analysis. The authors found that L-CNPs display more efficient performance than a conventional chemical fungicide like Ridomild Gold SL against pathogens such as Fusarium verticillioides. The most relevant outcomes found in the present work could have a positive impact on many industrial sectors, such as agriculture, drug-delivery, and the design of more green-friendly materials, among others. The achieved results are well-discussed during the main body of the reported manuscript. The scientific paper is well written. In my opinion the present manuscript is innovative and the methodological approached used matches with the scope of Polymers. For the above described reasons, I will recommend the publication in Polymers once the following remarks are fixed:
--------
KEYWORDS
(OPTIONAL) It may be desirable to introduce some additional keywords relevant to this work like “antifungal” or “plant growth”.
--------
INTRODUCTION
Introduction section is clear and concise. Some minor remarks must be addressed in order to increase the quality of the scientific content shown in the present manuscript:
“(…) 2.4% production yield (…) 7% yield (…)” (lines 49-50). Please, the authors should homogenize the significant figures. The authors should cover the same suggestion for the subsequent manuscript sections like the following sentence “wavelengths 452.5, 644, and 663 nm” (line 187) that appears in the M&M section.
“Additionally, several researchers have questioned the use of nanoparticles in plants to increase agricultural output due to their unfavorable effects on the environment and living things” (lines 79-81). Here, the authors should add the following reference citation [1].
[1] Ray, P.C.; et al. Toxicity and Environmental Risks of Nanomaterials: Challenges and Future Needs. J. Environ. Sci. Health C Environ. Carcinog. Ecotoxicol. Rev. 2009, 27, 1-35. https://doi.org/10.1080/10590500802708267.
--------
MATERIALS AND METHODS
Materials and methods employed by the authors are unequivocally described which is crucial to mimic the same experimental approach in other labs placed in different locations. Only the following remarks should be fixed:
“Lignin alkali was acquired from Sigma-Aldrich (St. Louis, MO)” (line 99). Please, the country of the supplier should be stated. Same comment for the following statements: “bovine serum albumin (BSA) (Sigma-Aldrich St. Louis, MO)” (lines 103-104), “2% dextrose; BD Difco, Sparks, MD)” (lines 156-157) and “Seed of maize (cv. Belgrano)” (line 162).
Furthermore, it lacks this type of information for these consumables used by the authors: “sterile phosphate-buffered saline (PBS) solution” (lines 159-160) and “Hematoxylin and eosin” (line 264).
Is the phosphate-buffered saline solution the same reagent as the reagent indicated in the subsection titled “Reagents”: “sodium phosphate buffer” (line 101)? In case affirmative, the authors should homogenize the terms.
“F.verticillioides” (line 110). Please, the authors should highlight this word in Italics. Same comment for lines 111, 117, 139, 140, 145, 148 and 149.
“(…) 25 mg L-1, 50 mg L-1, 100 mg L-1, and 500 mg L-1” (lines 164-165). Please, the authors should use the superscript for “L-1” (like in line 170). Same comment for “via CO2 (…)” (line 252), but in this case the subscript must be employed.
--------
RESULTS AND DISCUSSION
Authors perfectly state the most relevant outcomes found in the present work. Some points should be addressed to improve the manuscript quality.
I) “3.1. Synthesis and characterization of lignin-loaded carbon nanoparticles”. (line 274). Here, the manuscript will gain interest if the authors add a schematic representation of the chemical reactions taken place in the subsequent steps to functionalize the carbon nanoparticles with lignin. Furthermore, this information will aid to potential readers to better understand the strategy followed by the authors.
II) Then, the lignin alkali used by the authors contains nearby 4% of sulfur impurities. Could this aspect negative interfere during the functionalization of carbon nanoparticles with lignin? A brief explanation should be provided in this regard.
III) “3.1. Synthesis and characterization of lignin-loaded carbon nanoparticles”. (line 274) and “3.2. Molecular Identification of the fungal pathogen Fusarium verticillioides” (line 294). It lacks information of the used techniques in these subsections (scanning electron microscopy, Z-potential and PCR) in the respective Materials & Methods sections. Please, these details should be implemented.
IV) Figure 1, panel (B) (line 290). Could the authors increase the resolution of this Figure?
V) “Significant inhibition percentages of 91.23 ± 1.35, 95.23 ± 1.35, 96.22± 2.10, and 98± 1.25 %” (line 311). Please, the authors should homogenize the significant figures.
VI) “Furthermore, throughout the acid treatment operation, the number of peakds ascribed to the aldehyde decreased” (lines 365-366). Did the authors perform fourier transform infrared spectroscopy (FTIR) experiments? In case affirmative, this information should de detailed in Supplementary Information.
VII) Figure 9 (line 632). Please, scale bars should be inserted in all the panels that form this Figure.
VIII) “On the other hand, other carbon nanomaterials, (…) even though carbon is regarded as inert for biological systems, the potential toxicity of nanoparticles for human and animal health is still a major issue”. I agree with this statement provided by the authors and the following literature should be cited:
[2] Yuan, X.; et al. Cellular Toxicity and Immunological Effects of Carbon-based Nanomaterials. Part. Fibre Toxicol. 2019, 16, 18. https://doi.org/10.1186/s12989-019-0299-z.
--------
CONCLUSIONS
The authors should consider changing the current title by “Conclusions and Future perspectives” in order to cover the potential future avenues that the present research could pursue. In this context, it may be desirable to highlight some potential industrial applications derived in the use of lignin like the enhancement of antibacterial properties [3], the manufacturing of fertilizing products [4], the improvement wettability through the decrease of their mechanical properties [5] or for drug delivery purposes [6].
[3] Gerbin, E.; et al. Dual Antioxidant Properties and Organic Radical Stabilization in Cellulose Nanocomposite Films Functionalized by In Situ Polymerization of Coniferyl Alcohol. Biomacromolecules 2020, 21, 3163-3175. https://doi.org/10.1021/acs.biomac.0c00583.
[4] Savy, D.; et al. Novel fertilising products from lignin and its derivatives to enhance plant development and increase the sustainability of crop production. J. Clean. Prod. 2022, 366, 132832. https://doi.org/10.1016/j.jclepro.2022.132832.
[5] Marcuello, C.; et al. Atomic force microscopy reveals how relative humidity impacts the Young’s modulus of lignocellulosic polymers and their adhesion with cellulose nanocrystals at the nanoscale. Int. J. Biol. Macromol. 2020, 147, 1064-1075. https://doi.org/10.1016/j.ijbiomac.2019.10.074.
[6] Yiamsawas, D.; et al. Morphology-Controlled Synthesis of Lignin Nanocarriers for Drug Delivery and Carbon Materials. ACS Biomater. Sci. Eng. 2017, 3, 2375-2383. https://doi.org/10.1021/acsbiomaterials.7b00278.
--------
REFERENCES
Bibliography citations are in the proper format of Polymers. No further actions are requested for this section.
--------
OVERVIEW AND FINAL COMMENTS
The submitted work is well-designed and the gathered results are interesting to better understand the positive impact of lignin-loaded carbon nanoparticles on maize seed development and plant growth against Fusarium verticillioides microorganism. Furthermore, the present work provides a full analysis of the L-CNPs effect under different exposition times and particle concentrations. For these reasons, I will recommend the present scientific manuscript for further publication in Polymers once all the aforementioned suggestions will be properly fixed.
Reviewer 4 Report
The study developed a lignin-modified carbon nanoparticle as a seed coating agent and showed good results against the defence-related enzymes and disease resistance of Fusarium oxysporum. Thus, I am recommending it for publication after major revision.
1. Language of the manuscript needs substantial improvements. Authors should get ask someone to revise, and remove the errors, and improve the readability.
2. Do not use first person (we, our, us) in text.
3. The abstract is too long and not logical enough, and it is recommended that the abstract be streamlined.
4. Abstract,Line40,The article does not reflect that lignin-loaded carbon nanoparticles was cost-effective and environmentally for long term plant protection. Commercial alkaline lignin is not cheap to buy, while the preparation process is more complex, so how can it be used on a large scale in agriculture? Additional costing and environmental significance are recommended.
5. As a call for papers for Special Issue: Polymers: Environmental Aspects, this manuscript is more biased towards the application of lignin, with less research on the environmental role. It is recommended to highlight the significance of the environmental role.
6. Line99, should be expressed as “alkali lignin”, instead of “lignin alkali”.
7. Section 3.1. Synthesis and characterization of lignin-loaded carbon nanoparticles, Why does the black area in Figure 1A look like a pore structure? How can we explain the loading of alkali lignin on the surface of the carbon particles? It is also suggested to add more characterization of the physicochemical properties of lignin-loaded carbon nanoparticles, which take up too little space in the article.
8. What role does alkali lignin play as a modifier of carbon nanoparticles in disease resistance? Please provide a reasonable explanation.
9. Alkaline lignin itself is highly alkaline (pH>10) and can inhibit enzyme activity. Do charcoal particles modified with alkaline lignin as granules inhibit the germination process of maize seeds? Similarly, does good disease resistance result from the alkaline nature of alkaline lignin? Blank experiments with alkaline lignin only and carbon nanoparticles only should be added for clarification.
10. What are some important conclusions that can be drawn from this article through the study? The conclusions are not sufficiently represented in the abstract and conclusion sections and are suggested to be further improved.
Round 2
Reviewer 3 Report
The quality of the submitted manuscript has significantly improved since the authors have covered many of the Reviewers' request.
It may be desirable to add some reference citations (as indicated in my first reviewing report) of the potential future applications that lignin coated nanoparticles can display for Industry being thus, benefitial to society.
Reviewer 4 Report
After major revision, I believe it has met the criteria for publication and agree to accept this manuscript.